# Online In-Tube Solid-Phase Microextraction Coupled with Liquid Chromatography–Tandem Mass Spectrometry for Automated Analysis of Four Sulfated Steroid Metabolites in Saliva Samples

**DOI:** 10.3390/molecules27103225

**Published:** 2022-05-18

**Authors:** Hiroyuki Kataoka, Daiki Nakayama

**Affiliations:** School of Pharmacy, Shujitsu University, Nishigawara, Okayama 703-8516, Japan; s1916073se@shujitsu.jp

**Keywords:** sulfated steroid metabolites, saliva, online automated analysis, in-tube solid-phase microextraction (IT-SPME), liquid chromatography–tandem mass spectrometry (LC–MS/MS), stable isotope dilution

## Abstract

Accurate measurement of sulfated steroid metabolite concentrations can not only enable the elucidation of the mechanisms regulating steroid metabolism, but also lead to the diagnosis of various related diseases. The present study describes a simple and sensitive method for the simultaneous determination of four sulfated steroid metabolites in saliva, pregnenolone sulfate (PREGS), dehydroepiandrosterone sulfate (DHEAS), cortisol sulfate (CRTS), and 17β-estradiol-3-sulfate (E2S), by online coupling of in-tube solid-phase microextraction (IT-SPME) and stable isotope dilution liquid chromatography–tandem mass spectrometry (LC–MS/MS). These compounds were extracted and concentrated on Supel-Q PLOT capillary tubes by IT-SPME and separated and detected within 6 min by LC–MS/MS using an InertSustain swift C18 column and negative ion mode multiple reaction monitoring systems. These operations were fully automated by an online program. Calibration curves using their stable isotope-labeled internal standards showed good linearity in the range of 0.01–2 ng mL^−1^ for PREGS, DHEAS, and CRTS and of 0.05–10 ng mL^−1^ for E2S. The limits of detection (*S/N* = 3) of PREGS, DHEAS, CRTS, and E2S were 0.59, 0.30, 0.80, and 3.20 pg mL^−1^, respectively. Moreover, intraday and interday variations were lower than 11.1% (*n* = 5). The recoveries of these compounds from saliva samples were in the range of 86.6–112.9%. The developed method is highly sensitive and specific and can easily measure sulfated steroid metabolite concentrations in 50 μL saliva samples.

## 1. Introduction

Steroid hormones include glucocorticoids, which regulate carbohydrate and lipid metabolism and have inflammatory and immunosuppressive effects; mineralocorticoids, which regulate blood pressure through ionic equilibrium in body fluids; and sex hormones, which control reproductive functions and secondary sexual characteristics [1,2,3]. Starting from cholesterol, all of these compounds are biosynthesized in the adrenal cortex, gonads, and brain by various enzymes, but their biosynthetic and metabolic pathways are complex, and the molecular roles of these compounds have not been fully determined [1,2]. After their biosynthesis, these steroid hormones are metabolized by Phase I reactions involving oxidation and reduction and Phase II reactions involving conjugation [1,3].

Most steroids are sulfated by sulfotransferases (SULTs), which increase their solubility in aqueous solution and their excretion into urine. In addition, some of these sulfated steroids are desulfated by steroid sulfatases (STSs), with these steroids returning to their free forms [1,3]. This cycle plays an important role in the regulation of total steroid content and bioactivity in vivo [4,5], and the lack of STSs is a major factor in the pathogenesis of STS deficiency, recessive X-linked ichthyosis (RXLI), and the metabolic syndrome [6,7]. The ability to accurately measure the concentrations of sulfated steroid metabolites in vivo can not only provide insight into the regulation of steroid metabolism by the balance between SULTs and STSs, but also lead to the diagnosis of various diseases.

The sulfated steroid metabolite dehydroepiandrosterone-3-sulfate (DHEAS) [8,9] has been reported to be involved in responses to stress [10], reproductive function [5], the onset of age-related diseases, and human longevity [4,11]. During acute periods of stress, salivary DHEAS concentrations increase, but their levels decrease in long-term situations [10,12]. In addition, low DHEAS concentrations have been observed in subjects with aging-related diseases, such as sarcopenia, Alzheimer’s disease, depression, cardiovascular disease, and low libido [4,11].

Sulfated steroid metabolites can be analyzed by immunoassays or by mass spectrometry coupled to chromatography. Immunoassays, however, are not suitable for simultaneous analysis of a series of steroids due to their cross-reactivity, difficulties distinguishing among steroids with similar structures, and the need to generate specific antibodies against each compound [2,3,10,13]. Gas chromatography–mass spectrometry (GC–MS) cannot directly detect conjugated metabolites and requires chemical or enzymatic cleavage of sulfate groups and volatile derivatization steps [1,2,6,13,14,15]. Chemical cleavage of sulfated steroid metabolites also results in the hydrolysis of other conjugates, and commercially available sulfatase enzymes, which enzymatically cleave these sulfated compounds, also possess glucuronidase activity, making it difficult to distinguish between steroid glucuronides and sulfated steroids [6,14]. In contrast, liquid chromatography–mass spectrometry (LC–MS) and LC–tandem mass spectrometry (MS/MS) exhibit excellent ionization properties via electrospray ionization and do not require derivatization of compounds, and therefore the specific fragmentation patterns of these compounds allow selective and sensitive analysis by MS/MS [1,2,3,4,6,7,10,13,14,15,16,17,18,19,20,21,22,23,24,25].

Sulfated steroids have been measured in plasma, serum, and urine samples. However, blood collection is invasive and may itself induce stress, whereas urine collection is simple, but urinary concentrations of compounds are affected by the volume of urine excreted. In contrast, saliva can be easily collected non-invasively from subjects of all ages, from children to the elderly, and collection devices are relatively inexpensive. Furthermore, the concentrations of sulfated steroids in saliva are highly correlated with their concentrations in plasma or serum [10,23]. Because sulfated steroid content is lower in saliva than in serum, tedious and laborious pretreatment operations such as organic solvent extraction [23] and solid-phase extraction [10,20] are essential to separate and extract target analytes from the samples. 

In-tube solid-phase microextraction (IT-SPME) is a method by which samples can be easily extracted and concentrated using open-tube fused silica capillaries with coated inner surfaces as extraction devices, followed by online coupling to LC and LC–MS online using column switching technique (Appendix A). The entire process, from sample extraction/concentration to separation, detection, and data analysis, can be fully automated [26,27,28]. This method has been used to develop online analytical systems for a variety of compounds [28]. In addition, highly sensitive analytical methods were developed to determine the concentrations of non-sulfated steroid hormones in urine and saliva samples [29,30,31,32].

The aim of this study was to establish a fully automated online simultaneous analysis system, consisting of IT-SPME coupled with stable isotope dilution LC–MS/MS, for four sulfated steroid metabolites (Appendix A), pregnenolone sulfate (PREGS), DHEAS, 17β-estradiol 3-sulfate (E2S), and cortisol 21-sulfate (CRTS), which act as neuroactive steroids [4,8,11,33], and apply this system to the non-invasive analysis of these compounds in saliva samples.

## 2. Results and Discussion

### 2.1. Optimization of IT-SPME and Desorption of Sulfated Steroid Metabolites

An IT-SPME system that uses a capillary column as an extraction device involves the online extraction of the compounds of interest on the capillary column and the online desorption of these compounds by switching the draw/ejection flow of the sample solution and mobile phase flow (Appendix A). Extraction efficiency is mainly affected by the type of capillary coating, the number and flow rate of draw/eject cycles, and sample pH. Based on previous findings [31,32], these IT-SPME conditions were optimized for 1 ng mL^−1^ each of PREGS, DHEAS, and CRTS and 5 ng mL^−1^ of E2S. As PLOT columns including Supel-Q and Carboxen 1006 have a larger adsorption surface area and thicker film layer, the amounts extracted were greater than those for other liquid-phase columns (Figure 1A). Supel-Q has a higher affinity for sulfated steroids with cyclic skeletons than Carboxen 1006, which is a carbon molecular sieve, due to its divinylbenzene structure. All four sulfated steroid metabolites were efficiently extracted into a Supel-Q PLOT capillary by more than 25 repeated draw/eject cycles of 40 μL sample at a flow rate of 0.2 mL min^−1^ (Figure 1B). The length of the capillary is dependent on the draw/eject volume of the sample and is an important factor affecting extraction efficiency and time. However, capillaries that are too long and sample volumes that are too large will increase band width and require more time. Comparisons showed that, for a draw/eject volume of 40 µL of sample, a 60 cm long capillary with an inner diameter of 0.32 mm was optimal. In contrast, adjustment of sample pH can improve the distribution coefficient of compounds by suppressing their ionization. The optimal pH for non-sulfated steroid hormones in the previous reports [29,31,32] was 4, but PREGS, DHEAS, CRTS, and E2S contain sulfate groups, and therefore all four should be extracted into the capillary stationary phase at a more acidic pH. A pH that is too low, however, may cause damage to the extraction coating, affecting its service life and enrichment effect. Among the pH 2–9 buffers tested, potassium hydrogen phthalate–HCl buffer (pH 3) was found to be the most effective (Figure 2).

The absolute extractable amounts of sulfated steroid metabolites onto the capillary column were calculated by comparing peak area counts with the corresponding amounts in standard solution directly injected onto the LC columns. Using a 1 mL standard solution containing 1.0 ng mL^−1^ of each compound, the average extraction yields (*n* = 3) of PREGS, DHEAS, CRTS, and E2S onto the Supel-Q PLOT capillary column were found to be 56.6 ± 3.5%, 45.0 ± 2.2%, 47.3 ± 2.4%, and 39.7 ± 0.8%, respectively. The sulfated steroid metabolites extracted into the capillary column were dynamically desorbed and introduced directly into the LC column by online mobile phase flow using a column switching system. After analysis, the capillary column was cleaned and conditioned with methanol and mobile phase flow prior to the next analysis, thereby allowing its repeated use without carryover. All of these operations were programmed and automated (Appendix A).

### 2.2. LC–MS/MS Analysis of Sulfated Steroid Metabolites and Their Stable Isotope-Labeled Compounds

Four sulfated steroid metabolites, along with their stable isotope-labeled compounds as internal standards (ISs), exhibited abundant deprotonated ions [M−H]^−^ (Q1 mass) in the ESI-negative ionization mode. The [HSO_4_]^−^ ion (*m*/*z* 97) and [SO_3_]^−^ ion (*m*/*z* 80) formed by cleavage of the [M−H]^−^ of each compound were detected as fragment ions (Q3 mass). For each precursor ion [M−H]^−^, the *m*/*z* 97 for PREGS, DHEAS, CRTS, and their stable isotope-labeled compounds and the *m*/*z* 80 for E2S and E2S-d_4_ were selected as product ions, and the MS/MS operating parameters were optimized. The MRM transitions and MS/MS parameters set for each compound are shown in Appendix A. These findings were in good agreement with previously reported results [13,19,23].

The four sulfated steroid metabolites and their IS compounds were separated by LC on an InertSustain Swift C18 column. Chromatographic conditions were optimized by focusing on short retention times, paying special attention to matrix effects as well as peak shapes. Optimal separation was achieved using water/acetonitrile (55/45, *v*/*v*), with a flow rate of 0.2 mL min^−1^ resulting in good peak shapes and selective detection in MRM mode with a runtime of 6 min (Figure 3). The CV% of the retention time for each compound was within 5%. The analysis time per sample was about 23 min, allowing automated analysis of about 60 samples per day by operating overnight.

### 2.3. Analytical Method Validation and Advantages of IT-SPME LC–MS/MS Method

The analytical method was validated based on generally accepted validation criteria recommended in the ICH guidelines [34]. Linearity for PREGS, DHEAS and CRTS was validated by triplicate analyses of each compound at eight concentrations (0.01, 0.02, 0.05, 0.1, 0.2, 0.5, 1.0, and 2.0 ng mL^−1^), in the presence of 0.2 ng mL^−1^ each of PREGS-^13^C_2_-d_2_, DHEAS-d_5_, and CRTS-d_4_, respectively, whereas linearity for E2S was validated at concentrations of 0.05, 0.1, 0.2, 0.5, 1.0, 2.0, 5.0, and 10.0 ng mL^−1^ in the presence of 1.0 ng mL^−1^ E2S-d_4_. Calibration curves for each compound were linear, with correlation coefficients above 0.9998 (Table 1). The CVs of the peak height ratios at each concentration ranged from 0.7% to 12.6% (*n* = 3).

PREGS, DHEAS, CRTS, and E2S gave superior responses in MRM mode detection, with the LODs (*S/N* = 3) of each sulfated steroid metabolite in standard solutions ranging from 0.30 to 3.20 pg mL^−1^ (Table 1). The IT-SPME method was about 76-fold more sensitive than the direct injection method (10 µL injections).

Precision was assessed at concentrations of 0.05, 0.2, and 1.0 ng mL^−1^ for PREGS, DHEAS, and CRTS and of 0.25, 1.0, and 5.0 ng mL^−1^ for E2S. The precision, expressed as CV (%), was validated by performing five independent analyses on the same day and on five different days. The intraday and interday variations of these analyses were found to be 2.1–7.7% and 4.0–11.1%, respectively (Table 2).

These results obtained based on the ICH guidelines showed that the IT-SPME LC−MS/MS method has good linearity and precision. The method is simpler, more sensitive, and more specific than previously reported methods [6,7,16,17,18,19,20,21,25] and can be fully automated from extraction and concentration of the four sulfated steroid metabolites to their separation and analysis.

### 2.4. Application to the Analysis of Saliva Samples

Because the concentrations of steroid hormones and their metabolites in saliva have been reported to reflect their free concentrations in plasma or serum [10,23], saliva is regarded as an excellent physiological medium for non-invasive sampling [10,35,36]. Therefore, saliva analysis can allow, in particular cases (i.e., patients with difficult blood collection), the replacement of blood analysis [36]. In general, however, procedures for the collection, handling, and storage of salivary samples should be considered, because the levels of biomarkers in saliva are affected by factors such as sex, age, smoking, diet, circadian rhythm, and more [10,35,36]. The most commonly used saliva collection procedures are passive drooling, the use of cotton swabs, aspiration through soft devices positioned under the tongue, and chewing of paraffin gum [35,36], but the choice of sampling tube type (glass or polystyrene) has no effect on salivary steroid concentration [10]. Since some biomarkers in saliva are unstable compounds, they need to be cooled after collection and frozen if not analyzed immediately [10,35,36]. In this study, saliva samples were collected into Salisoft tubes containing polypropylene–polyethylene swab, followed by ultrafiltration with Amicon Ultra to remove high-molecular-weight components in saliva, such as mucins and other coexisting proteins. The polymeric saliva collectors, such as Salisoft, have a better recovery rate of steroid hormones compared to cotton saliva collectors [10,32,37]. Furthermore, the 30K (molecular weight 30,000 cutoff) centrifugal filter used for ultrafiltration of saliva samples in the previous report [32] was replaced with a 3K (molecular weight 3000 cutoff) filter to reduce matrix effects.

Stable isotope-labeled compounds as ISs were added to saliva samples prior to extraction to correct the influence of matrix effects on the analysis of sulfated steroid metabolites in the samples. As shown in Figure 4, the saliva samples were successfully analyzed without interference peaks by the IT-SPME LC−MS/MS method with MRM mode detection. The LOQs (*S/N* = 10) of these sulfated steroid metabolites ranged from 16 to 172 pg mL^−1^ saliva (Table 1). In comparison, the LOQs for DHEAS by previously reported LC−MS/MS methods ranged from 0.06 to 1.14 ng mL^−1^ saliva [20,23], indicating that the IT-SPME LC−MS/MS method was 3.7 times more sensitive than these methods.

To confirm the validity and accuracy of this method, known amounts of PREGS, DHEAS, CRTS, and E2S were spiked into saliva samples, and their recoveries were calculated. The overall recoveries of PREGS, DHEAS, CRTS, and E2S were above 86%, with relative standard deviations of 0.8–9.7% (Table 3). These results show that the IT-SPME LC−MS/MS method has good accuracy and precision and is fully applicable to saliva samples.

The developed method was used to analyze the concentrations of the four sulfated steroid metabolites in saliva samples from 10 male and 10 female subjects. PREGS and DHEAS were detected in all saliva samples, with DHEAS being present at relatively high concentrations (Table 4). In contrast, CRTS and E2S were often below the LOQ because of interference by coexisting peaks, even if peaks were detected. Salivary DHEAS concentrations ranged from 0.36 to 11.9 ng mL^−1^ in males and from 0.05 to 4.8 ng mL^−1^ in females, with lower concentrations in children than in adults. These results are similar to findings showing that DHEAS concentrations peak at around ages 20 to 30 years in both men and women and decrease subsequently with age [4,6].

## 3. Materials and Methods

### 3.1. Reagents and Standard Solutions

PREGS sodium salt and E2S sodium salt were purchased from Sigma-Aldrich Japan (Tokyo, Japan), DHEAS hydrate was from Tokyo Kasei Kogyo (Tokyo, Japan), and CRTS potassium salt was from Toronto Research Chemicals Inc. (TRC, North York, ON, Canada). Their stable isotope-labeled compounds, PREGS-^13^C_2_-d_2_ (isotopic purity >98%, Sigma-Aldrich, St. Louis, MA, USA), DHEAS-d_5_ (isotopic purity >90%, Sigma-Aldrich), CRTS-d_4_ (isotopic purity >95%, TRC), and E2S-d_4_ (isotopic purity 95%, TRC), were used as internal standards (IS). These standard and IS compounds (Appendix A) were dissolved in LC–MS-grade methanol to a concentration of 0.1 mg mL^−1^ and diluted with LC–MS-grade distilled water to the required concentration prior to use. The mixed standard solution consisted of 20 ng mL^−1^ each of PREGS, DHEAS, and CRTS and 100 ng mL^−1^ E2S, whereas the mixed IS solution consisted of 2 ng mL^−1^ each of PREGS-^13^C_2_-d_2_, DHEAS-d_5_, and CRTS-d_4_ and 10 ng mL^−1^ of E2S-d_4_. All of these solutions were stored at 4 °C. LC−MS-grade acetonitrile and distilled water used as mobile phases were purchased from Kanto Chemical (Tokyo, Japan); all other chemicals were of analytical reagent grade.

### 3.2. Preparation of Saliva Samples

Saliva samples were obtained from 20 healthy volunteers (10 men and 10 women). The experimental protocol was approved by the ethics committee of Shujitsu University (approval code 207; 14 October 2020), and all volunteers provided written informed consent. Saliva samples were collected in Salisoft tubes (Assist, Tokyo, Japan). The tubes were centrifuged at 2500× *g* for 1 min to elute the saliva solution. If not immediately used for analysis, the samples were stored frozen at −20 °C and thawed spontaneously just before analysis. A 0.1 mL aliquot of mixed IS solution was added to 0.05 mL of each saliva sample, followed by ultrafiltration using an Amicon Ultra 0.5 mL 3K (Millipore, Tullagreen, Ireland) regenerated cellulose 3000 molecular weight cutoff centrifugal filter device at 15,000 rpm for 20 min. Each filtrate was pipetted into a 2.0 mL autosampler vial with septa, to which was added 0.05 mL of 0.2 M potassium hydrogen phthalate–HCl buffer (pH 3). The total volume was made up to 1.0 mL with distilled water, and the vials were set into the autosampler for IT-SPME LC−MS/MS analysis. The concentrations of the sulfated steroid metabolites in saliva were calculated using calibration curves constructed from the ratios of the peak heights of each sulfated steroid metabolite to the peak height of their IS compounds.

### 3.3. LC−MS/MS Analysis

LC–MS/MS analysis was performed using an Agilent Technologies (Boeblingen, Germany) Model 1100 series LC system and an Applied Biosystems (Foster City, CA, USA) API 4000 triple quadrupole mass spectrometer, with LC separation on an InertSustain swift C18 column (100 mm × 2.1 mm, particle size 5 μm; GL Sciences, Tokyo, Japan). The LC conditions included a column temperature of 30 °C, a mobile phase consisting of distilled water/acetonitrile (55/45, *v*/*v*), and a flow rate of 0.2 mL min^−1^. Electrospray ionization (ESI)–MS/MS conditions included: a turbo ion spray voltage of –4500 V; a turbo ion spray temperature of 450 °C; ion source gas GS1 and GS2 flows of 20 and 11 L min^−1^, respectively; a curtain gas (CUR) flow of 10 L mL^−1^; and a collision gas (CAD) flow of 4.0 L min^−1^. Multiple reaction monitoring (MRM) transitions in negative ion mode and other parameters, including dwell time, declustering potential (DP), entrance potential (EP), collision energy (CE), and collision cell exit potential (CXP), are shown in Appendix A. Quantification was performed by MRM of the deprotonated precursor molecular ions [M−H]^−^ and the related product ions for each compound. Quadrupoles Q1 and Q3 were set at unit resolution (Appendix A). Analyst Software 1.6.2 (Applied Biosystems) was used for LC–MS/MS data analysis.

### 3.4. In-Tube SPME

IT-SPME was essentially performed as described in our previous works [31,32]. A GC capillary column (60 cm × 0.32 mm i.d.) as an extraction device was connected between the injection needle and injection loop of the autosampler. The capillary column was threaded through a 1/16 inch polyetheretherketone (PEEK) tube with a length of 2.5 cm long and an inner diameter of 330 μm and connected using standard 1/16 inch stainless steel nuts, ferrules, and connectors. CP-Sil 5CB (100% polydimethylsiloxane, film thickness 5 μm), CP-Sil 19CB (14% cyanopropyl phenyl methylsilicone, film thickness 1.2 μm) (Varian Inc., Lake Forest, CA, USA), Supelco-Wax (polyethylene glycol, film thickness 1.0 μm), Supel-Q PLOT (divinylbenzene polymer, film thickness 17 μm), and Carboxen 1006 PLOT (carbon molecular sieve, film thickness 15 μm) (Supelco, Bellefonte, PA, USA) were used to compare extraction efficiencies. Extraction, desorption, and injection parameters were programmed by the autosampler software (Appendix A) [31,32].

## 4. Conclusions

In this study, we succeeded for the first time in efficiently extracting and concentrating highly polar sulfate conjugates by IT-SPME at acidic pH, and we constructed an automated analysis system coupled online with LC–MS/MS to enable selective and sensitive simultaneous analysis of four sulfated steroid metabolites. The method is easy to apply to the analysis of small volumes of saliva samples without tedious pretreatment except for ultrafiltration. This method may be a useful tool in analyzing the regulation of steroid metabolism and in determining the diagnosis of related diseases.

## Figures and Tables

**Figure 1 molecules-27-03225-f001:**
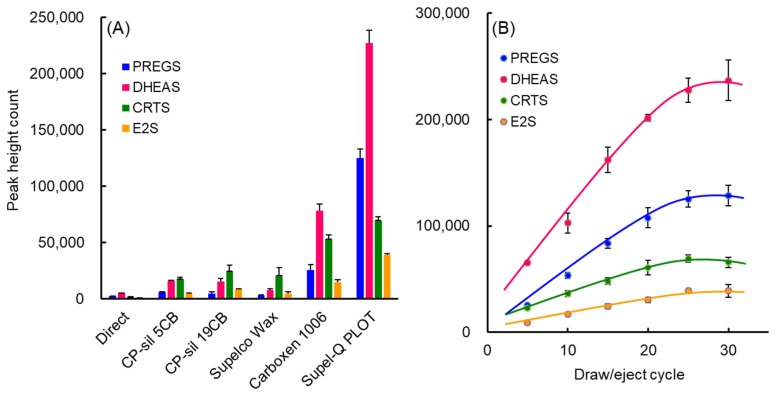
Effects of (**A**) capillary coatings and (**B**) number of draw/eject cycles on the IT-SPME of sulfated steroid metabolites. Standard solution containing 1 ng mL^−1^ each of PREGS, DHEAS, and CRTS and 5 ng mL^−1^ of E2S were extracted by (**A**) 25 draw/eject cycles of 40 µL of standard solution at a flow rate of 200 µL min^−1^, and (**B**) the indicated number of draw/eject cycles of 40 µL of standard solution on a Supel-Q PLOT capillary at a flow rate of 200 µL min^−1^.

**Figure 2 molecules-27-03225-f002:**
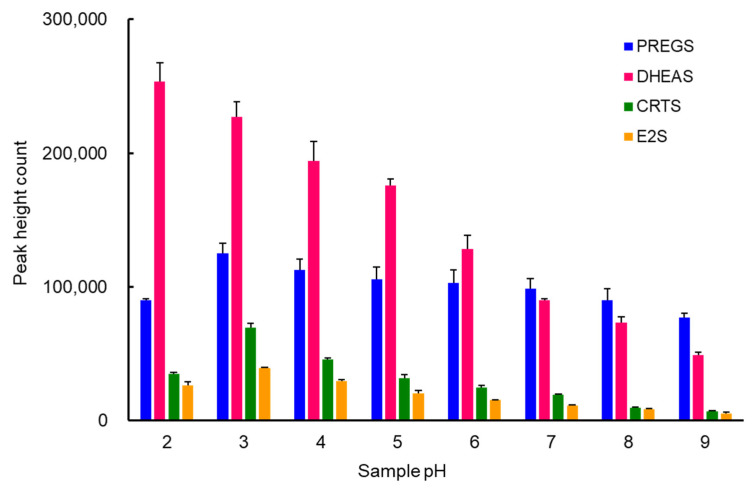
Effects of sample pH on the IT-SPME of sulfated steroid metabolites. Standard solution (40 µL) containing 1 ng mL^−1^ each of PREGS, DHEAS, and CRTS and 5 ng mL^−1^ of E2S were extracted by 25 draw/eject cycles on a Supel-Q PLOT capillary at a flow rate of 200 µL min^−1^.

**Figure 3 molecules-27-03225-f003:**
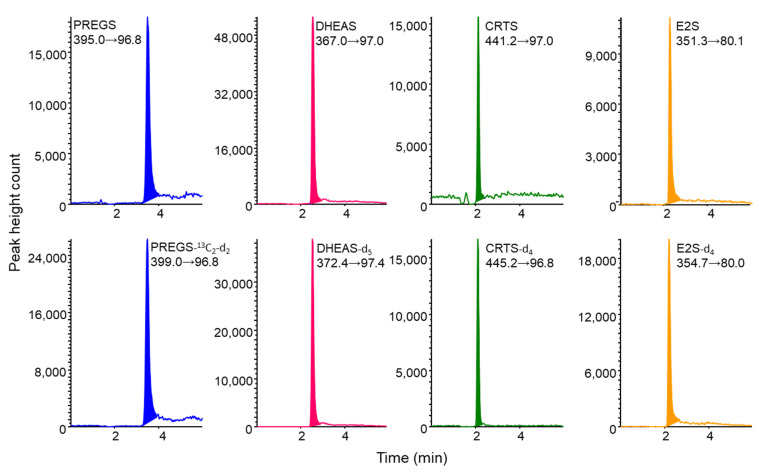
Multiple reaction monitoring (MRM) chromatograms obtained from standard solution containing 0.2 ng mL^−1^ each of PREGS, DHEAS, and CRTS and 1 ng mL^−1^ of E2S, and their stable isotope-labeled compounds. IT-SPME LC–MS/MS conditions are described in Section 3.

**Figure 4 molecules-27-03225-f004:**
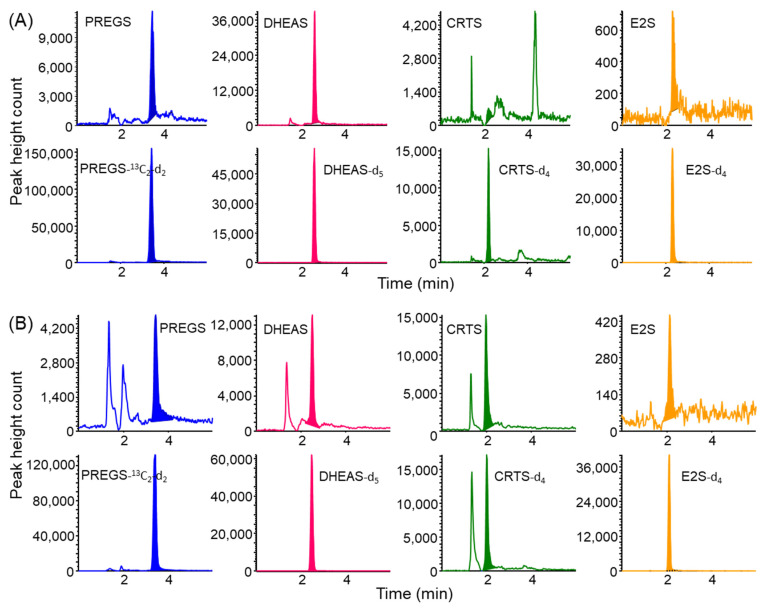
MRM chromatograms obtained from saliva samples of (**A**) a non-smoker and (**B**) a smoker by IT-SPME LC–MS/MS. Analytical conditions are described in Section 3.

**Table 1 molecules-27-03225-t001:** Linearity and sensitivity of the IT-SPME LC–MS/MS method for sulfated steroid metabolites.

Compound	Linearity		LOD ^2^ (pg mL^−1^)		LOQ ^3^ (pg mL^−1^)
	Range (ng mL^−1^)	CC ^1^	Direct Injection	IT-SPME	IT-SPME
PREGS	0.01–2	0.99991	50.6	0.59	28
DHEAS	0.01–2	0.99994	23.9	0.30	16
CRTS	0.01–2	0.99987	68.4	0.80	47
E2S	0.05–10	0.99995	245.4	3.20	172

^1^ Correlation coefficient (*n* = 24); ^2^ limits of detection: pg mL^−1^ sample solution (signal-to-noise ratio of 3); ^3^ limits of quantification: pg mL^−1^ saliva sample (signal-to-noise ratio of 10).

**Table 2 molecules-27-03225-t002:** Precision of the IT-SPME LC–MS/MS method for sulfated steroid metabolites.

Compound	Concentration (ng mL^−1^)	Precision (CV ^1^ %), (*n* = 5)	
		Intra-Day	Inter-Day
PREGS	0.05	2.3	9.0
	0.2	2.1	4.0
	1	3.0	6.2
DHEAS	0.05	6.9	10.7
	0.2	3.1	7.8
	1	3.1	6.1
CRTS	0.05	7.7	11.1
	0.2	2.4	4.2
	1	2.6	6.8
E2S	0.25	5.6	7.7
	1	3.6	7.9
	5	2.7	5.2

^1^ CV, coefficient variation.

**Table 3 molecules-27-03225-t003:** Recoveries of sulfated steroid metabolites spiked into saliva samples.

Compound	Spiked (ng mL^−1^ Saliva)	Recovery ± SD (%), (*n* = 3)
PREGS	1.0	87.6 ± 5.3
	4.0	90.1 ± 4.6
	20	112.9 ± 5.8
DHEAS	1.0	86.3 ± 2.4
	4.0	91.4 ± 5.2
	20	93.0 ± 0.7
CRTS	1.0	98.7 ± 9.5
	4.0	96.5 ± 7.0
	20	98.9 ± 3.4
E2S	5.0	86.6 ± 3.8
	20	105.5 ± 2.7
	100	106.5 ± 5.3

**Table 4 molecules-27-03225-t004:** Contents of sulfated steroid metabolites in saliva samples.

Subject			Content (pg mL^−1^ Saliva), (*n* = 3)			
No.	Sex ^1^	Age	PREGS	DHEAS	CRTS	E2S
1	M	6	45 ± 6	1068 ± 79	<LOQ	<LOQ
2	M	7	33 ± 1	869 ± 9	<LOQ	<LOQ
3	M	23	170 ± 14	1914 ± 146	187 ± 15	509 ± 1
4	M	24	128 ± 2	3894 ± 229	114 ± 5	306 ± 7
5	M	25	149 ± 15	5139 ± 71	314 ± 10	466 ± 12
6	M	35	48 ± 6	8272 ± 334	<LOQ	<LOQ
7	M	38	52 ± 2	6022 ± 25	<LOQ	<LOQ
8	M	40	72 ± 1	11,908 ± 730	873 ± 37	<LOQ
9	M	57	86 ± 1	570 ± 35	3215 ± 306	276 ± 21
10	M	67	32 ± 3	365 ± 38	<LOQ	174 ± 12
11	F	4	40 ± 3	47 ± 2	295 ± 18	184 ± 32
12	F	6	44 ± 3	69 ± 5	369 ± 14	179 ± 30
13	F	27	64 ± 0	1729 ± 85	<LOQ	<LOQ
14	F	29	109 ± 19	1244 ± 75	<LOQ	174 ± 13
15	F	30	46 ± 3	4415 ± 8	<LOQ	<LOQ
16	F	33	44 ± 2	4783 ± 120	200 ± 12	<LOQ
17	F	34	41 ± 4	129 ± 8	<LOQ	175 ± 9
18	F	36	90 ± 4	4607 ± 73	<LOQ	<LOQ
19	F	62	46 ± 1	1418 ± 26	<LOQ	<LOQ
20	F	64	28 ± 2	1217 ± 102	<LOQ	<LOQ

^1^ M, male; F, female.

## Data Availability

The data presented in this study are available on request from the corresponding author.

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
