# Peer review of "Online In-Tube Solid-Phase Microextraction Coupled with Liquid Chromatography–Tandem Mass Spectrometry for Automated Analysis of Four Sulfated Steroid Metabolites in Saliva Samples"

_molecules, 2022, doi:10.3390/molecules27103225_

Round 1

Reviewer 1 Report

Dear Authors, the method describes an interesting approach based on IT-SPME-LC-MS/MS for the determination of sulphated steroid metabolites in saliva. I have some suggestions to be included in the manuscript before to accept it in Molecules.

  1. I strongly suggest to compare your method with those already reported in literature highliting the main advantages of IT approach.
  2. did you observe any drawback of IT using real saliva? Did you filter saliva samples before to load it into IT?
  3. did you observe any carry-over effect of the IT device?
  4. did you evaluate the analyse stability once them are loaded into the sorbent material?
  5. What is the main difference between It and MEPS? Please discuss it in the manuscript.
  6. I strongly suggest to include the main advantages/disadvatange of saliva sampling. In particular attention should be focused on the impact of collection procedures on the concentration of the target analyses. You could refer to the following articles: 10.1016/j.trac.2019.115781, 10.1371/journal.pone.0114430, 10.1016/j.microc.2017.02.032

Author Response

The author's response to the reviewer report (Reviewer 1) is attached.

Reviewer 2 Report

The publication is well written, thoughtful and certainly interesting. I have several comments to the authors, one of which is fundamental: what is new in this work, what have the authors done here for the first time? Throughout the text, the authors refer to their previous results, refer to previous studies with citations. So I am beginning to wonder about the absolute novelty of this research. If the authors can clearly indicate novelty I will be in favour of accepting the publication, otherwise I may request  for rejection.

Other minor comments:

It is necessary to provide data for bioethics committee approval.

Figure 1A: please give a full interpretation of the results obtained here, why the values are so different depending on the capillary used? What is on their surface, what processes take place there? Do lower values mean lower adsorption or desorption (with full adsorption)?

Please give standard deviations for recoveries (lines 147-150)

Lines 147-150 and 312-316: what has changed that the recoveries have increased by a factor of two for real samples?

Author Response

The author's response to the reviewer report (Reviewer 2) is attached.

Round 2

Reviewer 1 Report

Dear Authors, all the questions were answered so I suggest to accept the manuscript in Molecules.